# EdgeThemis: Ensuring Model Integrity for Edge Intelligence

## ABSTRACT

Machine learning (ML) models are widely deployed on edge nodes, such as mobile phones and edge servers, to power a wide range of AI applications over the web. Ensuring the integrity of these edge models is paramount, as they are subject to corruption caused by software/hardware exceptions and malicious tampering, which may undermine model performance, incur economic losses, and pose health risks. Existing data integrity mechanisms designed for files stored on disks cannot properly verify the integrity of models running in GPUs or mitigate the new integrity threats against edge models. This paper proposes EdgeThemis, a novel mechanism for verifying the integrity of edge models through sentinel verification. To enable verifiability for a model $M$, EdgeThemis embeds a sentinel backdoor and a verification module into $M$. Then, a challenger can send verification requests to the edge node hosting $M$ to verify its integrity. Next, the sentinel activates the verification module to generate a unique integrity proof tied to the identity of the edge node for verification. Finally, the challenger can verify the integrity proof to detect model corruption. Theoretical analysis proves that EdgeThemis can properly mitigate potential integrity threats against edge models. Experiments demonstrate that EdgeThemis achieves a verification accuracy of 100.00% across various models and different types of model corruption with robustness against replay attacks, theft attacks, and replacement attacks.

## CCS CONCEPTS

• **Do Not Use This Code → Generate the Correct Terms for Your Paper**; *Generate the Correct Terms for Your Paper*; Generate the Correct Terms for Your Paper; Generate the Correct Terms for Your Paper.

## KEYWORDS

Model integrity, edge intelligence, edge computing

**ACM Reference Format:**
Anonymous Author(s). 2018. EdgeThemis: Ensuring Model Integrity for Edge Intelligence. In *Proceedings of Make sure to enter the correct conference title from your rights confirmation emai (Conference acronym 'XX)*. ACM, New York, NY, USA, 11 pages. https://doi.org/XXXXXXX.XXXXXXX

## 1 INTRODUCTION

Edge computing has become a crucial technology in modern web systems, enabling low-latency data processing and retrieval by bringing computation and storage resources closer to users [25, 38],

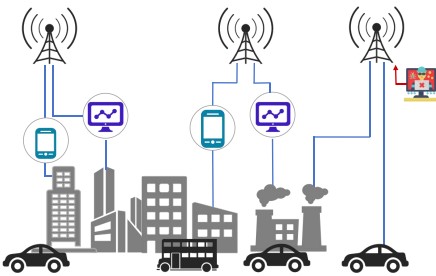

**Figure 1: Edge Computing System. ML models can be deployed on edge nodes like mobile devices and edge servers to offer users low-latency inference services. These models are subject to accidental and malicious corruption.**

as shown in Figure 1. This significantly reduces the latency associated with transmitting data to users, making it ideal for web applications [36] that demand fast responses. Deploying machine learning (ML) models on edge nodes like mobile devices and edge servers is another typical solution to improving web users' experience with mobile and web-of-things applications such as visual assistance [22] and video analytics [15]. These edge models excel at extracting information from complex data and tasks, enabling rapid decision-making [27]. A series of lightweight ML models have been designed to pursue edge-friendly ML models that can be deployed on edge nodes, such as MobileViT [26] and Tiny BERT [13].

However, the distributed nature of edge computing presents new and significant security challenges [9]. ML models encapsulate not only valuable intellectual property but also process sensitive information risking breaches and exploitation [10]. A model compromised due to software vulnerabilities [19], hardware failures [20], or malicious tampering such as backdoor attacks [8] or poisoning attacks [4] may perform incorrect inferences that cause significant harm to the model owner and/or its users [33]. For example, in autonomous driving, if a model controlling or coordinating a self-driving car's perception system is compromised, it may misclassify a stop sign as a speed limit sign, leading to severe traffic hazards and potentially endangering human lives [6, 23]. Therefore, mechanisms for verifying the integrity of edge models are vital and imminent.

Traditional data integrity verification methods are effective for verifying files stored in cloud nodes [3, 14] or edge nodes [9, 20]. Edge models are normally loaded into GPUs for inferences. They are susceptible to potential runtime attacks, such as memory tampering or model replacement (§2), which cripple traditional mechanisms for file integrity verification because they cannot effectively and efficiently detect runtime alterations in edge models running in GPUs. While file integrity verification checks the integrity of files saved on disks, models loaded into GPU can be altered or replaced with malicious versions. Thus, verifying the integrity of edge models at runtime is essential to ensure that the models serving users remain corruption-free.

Some research efforts have focused on verifying the integrity of models deployed in the cloud [11, 37] by selecting sensitive data based on models' decision boundaries and validating the correctness of their inference results. However, these methods cannot ensure that an edge model running in a GPU has not been altered accidentally or maliciously. Moreover, relying on decision boundaries limits the applicability of verification to specific types of models. This limitation renders these methods impractical because modern ML tasks encompass a wide range of AI applications that demand various types of models.

Running on edge nodes, edge models face new integrity threats. The edge node can use old integrity proofs to perform a replay attack. Alternatively, it can steal integrity proof from other edge nodes to cheat the verification process with a theft attack. It may also reload the original uncorrupted model from the model file to produce the integrity proof and cheat the verification process with a replacement attack.

**Objectives.** To tackle the abovementioned challenges, integrity verification for edge models must achieve four main properties.

- **Consistency.** The verification mechanism must be able to detect the inconsistency between an edge model and its original deployment, in terms of both model structure and model parameters.
- **Generality.** The verification mechanism must be able to accommodate different types of AI tasks and model architectures.
- **In-Memory.** The verification mechanism must be able to detect model corruption at runtime without model saved onto disks.
- **Security.** The verification mechanism must be robust against attempts by adversaries to cheat the verification process.

**Contributions.** This paper presents EdgeThemis, a new mechanism for verifying the integrity of edge models with sentinel verification. It makes the following main contributions.

- Before the deployment of an edge model on an edge node, EdgeThemis embeds a sentinel backdoor to the model through fine-tuning with a sentinel dataset, and integrates a verification module after the output layer. At runtime, a challenger can send a verification request that impersonates a normal request to trigger the sentinel backdoor with an irregular but smooth data sample. The sentinel will then activate the verification module covertly to prevent proactive integrity attacks.
- EdgeThemis utilizes a digest-based method for model verification. In response to a verification challenge, the edge node computes a digest based on the in-CPU model structure and in-GPU model parameters as the integrity proof. The digest differs from the static digest computed from the model file. It is returned to the challenger for validation.
- EdgeThemis ensures that an integrity proof is generated specifically for the current verification to mitigate replay attacks. In addition, it binds every integrity proof to the identity of the corresponding edge node to fight theft attacks. Finally, it sets an adaptive proof return timer to defend against replacement attacks under Byzantine settings.
- We analyze the properties of EdgeThemis theoretically and evaluate it experimentally against four baselines with five popular edge-friendly models[1]. The experimental results demonstrate

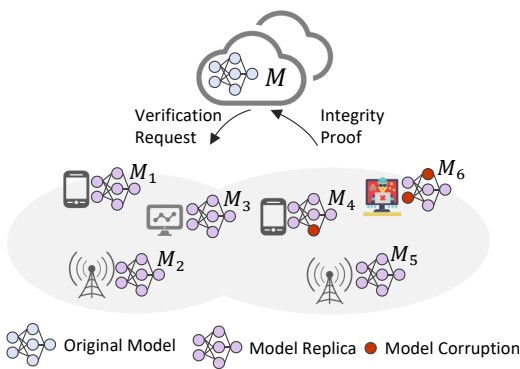

**Figure 2: Challenge-response Integrity Verification. The challenger challenges edge nodes who respond with integrity proofs.**

that EdgeThemis outperforms all baselines significantly, achieving a verification accuracy of 100.00%. It can detect different types of corruption to different degrees with robustness against replay, theft, and replacement attacks.

## 2 THREAT MODEL

Edge model integrity verification aims to ensure that the models running on edge nodes remain consistent with the original deployments, preventing model corruption caused by accidental corruption or malicious tampering. This threat model outlines potential attacks that could compromise edge model integrity, including adversary capabilities, corruption types, and malicious behaviors.

**Adversary Capabilities.** An adversary could be the edge node running the model or another edge node controlled by an external adversary. It can access the edge node's disks and memory to tamper with the model, such as modifying its parameters or structures, embedding backdoors, and compressing the model to save on storage resources. These adversaries may have various motivations, including access to sensitive information, financial gains, and service disruptions.

**Corruption Types.** Model corruption can be categorized into two types.

- Accidental Corruption. Model corruptions may be caused by hardware malfunctions, software anomalies, network disruptions, as well as other exceptions. Such corruption can lead to compromised model performance, inaccurate inference results, or complete model failures. As a result, users may suffer poor service quality and service disruptions.
- Malicious Tampering. Adversaries may alter model structure, modify model parameters, embed malicious behaviors, or replace the model with a fake or an inferior one, aiming to elicit incorrect inference results or harm the interests of the model owner.

**Malicious Behaviors.**

- Verification Detection. An adversary will try to detect verification so that it can launch an integrity attack to cheat the verification process.
- Replay Attack. The adversary may use prior integrity proofs that have been successfully verified to hide the fact that the edge model has been tampered with.

---

[1]The source code used in the evaluation is available at: https://anonymous.4open.science/r/EdgeThemis/.

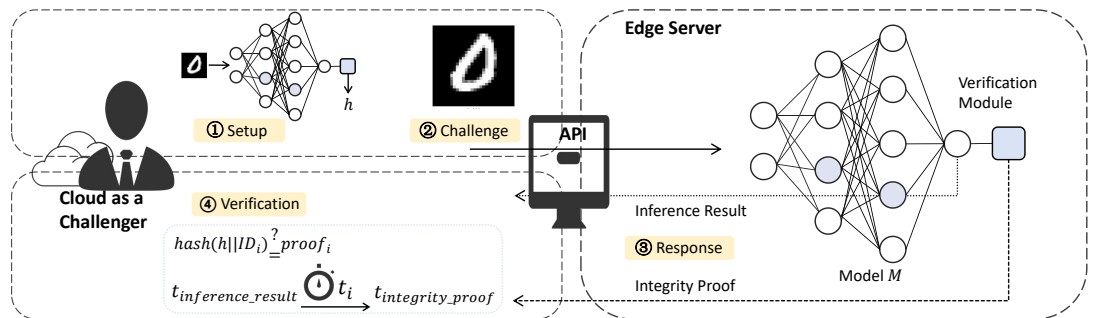

**Figure 3: Verification Process: The challenger sends a covert verification request to each edge node, activating its verification module through the sentinel backdoor to produce an integrity proof to be returned for validation.**

- **Theft Attack.** The adversary may steal an integrity proof from another edge node and use it as its own to conceal the corruption of its own model.
- **Replacement Attack.** The adversary may store the correct model file on disks and reload it into the GPU for proof generation when it is challenged to cheat the verification.

## 3 SYSTEM DESIGN

### 3.1 Overview

Before deploying a model $M$, the challenger embeds a sentinel backdoor (§3.2) and a verification module (§3.3) in $M$ after its output layer. After that, $M$ can be deployed on edge nodes to enable AI services.

At runtime, the challenger challenges the edge nodes to verify the integrity of their models periodically or on demand. As illustrated in Figure 2, EdgeThemis uses a challenge-response scheme to verify edge model integrity, following similar mechanisms for conventional integrity mechanisms [3, 14, 20]. The verification process goes through four steps, as illustrated in Figure 3.

① **Setup.** The cloud server, as a challenger, generates a verification sample $s$ with a decoder specifically trained for $M$ (§3.2).

② **Challenge.** The challenger sends $s$ as a covert verification request, among normal user requests, to a set of edge nodes to verify the integrity of their models without revealing its identity and purpose.

③ **Response.** Each edge node runs the verification request through the edge model to produce the inference result ($\hat{y}$), which will be returned to the challenger. In the meantime, the verification model (§3.3) is activated to produce an integrity proof, also to be returned to the challenger.

④ **Validation.** Upon receiving an integrity proof $proof_i$ from an edge node within a specific period of time (§3.4), the challenger validates its correctness against $h$, the digest of its own $M$:

$$hash(h||ID_i) \stackrel{?}{=} proof_i. \tag{1}$$

where $ID_i$ is the ID of the edge node.

### 3.2 Sentinel Embedding

EdgeThemis embeds a sentinel in a model to covertly activate the verification module (§3.3) when a verification request from the challenger comes through the sentinel backdoor. In response to normal user requests, the verification model remains inactive. This ensures that the challenger, aware of the hidden backdoor, can verify the model without revealing its identity and purpose. A simple way to achieve this objective is through backdoor embedding [1]. However, traditional backdoor methods, which rely on visible triggers like specific objects (e.g., eyeglasses in face recognition [8]), have drawbacks. First, when an adversarial edge node detects a trigger, it can coordinate an attack in response to cheat the verification. Second, legitimate user requests might unintentionally activate the verification module, compromising model performance and disrupting its inferences. To tackle these challenges, EdgeThemis employs a new method that embeds undetectable sentinel backdoors into models without impacting their responses to normal user requests.

**Step 1: Sentinel Dataset Generation.** Based on the above analysis, to ensure that verification samples resemble normal user requests, we need a dataset with a smooth appearance with irregular characteristics. This dataset will be used to fine-tune the original deployment model in order to embed the sentinel backdoor. We propose a method based on identifying hidden pathways within the model, as illustrated in Figure 4 and described in Algorithm 1 (see Appendix A).

EdgeThemis trains an autoencoder on the dataset used to train $M$, denoted by $D_{training}$. Its encoder is used to map $D_{training}$ into a latent space, forming an area of training data feature vectors $O_{training}$. In this space, EdgeThemis identifies $O_{sentinel}$, an outlier area that is sufficiently distant from and non-overlapping with $O_{training}$. EdgeThemis randomly selects $n$ points from $O$ and passes them through the decoder to generate a new dataset denoted by $D_{sentinel}$. This dataset is used to fine-tune $M$, embedding the sentinel and creating a backdoor for verification activation.

**Step 2: Model Fine-tuning.** The next step is to embed the sentinel backdoor into $M$ through a fine-tuning process. After that, $M$ must activate its verification module (§3.3) only in response to verification requests and not normal user requests. The labels of the samples $D_{sentinel}$, denoted by $y_{sentinel}$, should correspond to the outputs that normal samples cannot produce. This unique label is critical for distinguishing verification requests from normal user requests, ensuring that only verification requests will activate the verification module in $M$.

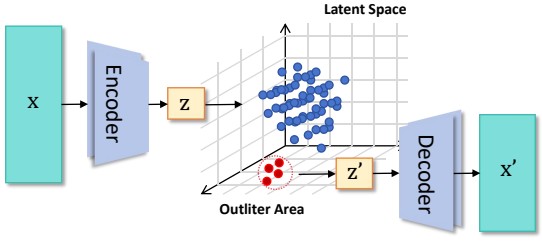

**Figure 4: Sentinel Dataset Generation. EdgeThemis can fine-tune a model on a specifically-generated sentinel dataset to enable its verifiability.**

Let $M_\theta$ denote the original model for deployment on edge nodes with parameters $\theta$ trained on the original dataset $D_{training}$. EdgeThemis fine-tunes $M_\theta$ on $D_{sentinel}$ to obtain a new set of parameters $\theta'$, such that:

$$\theta' = \arg\min_\theta \mathcal{L}(M_\theta(x_{sentinel}), y_{sentinel}), \qquad (2)$$

where $x_{sentinel}$ and $y_{sentinel}$ are the input data and corresponding labels from $D_{sentinel}$, and $\mathcal{L}$ is the loss function.

This fine-tuning process ensures that the sentinel data $x_{sentinel}$ activates the verification module, while label $y_{sentinel}$ serves as the condition for triggering the module.

### 3.3 Verification Module

In response to a verification request, $M$ first produces the result $\hat{y}$. In the meantime, the features $F$ produced by its last layer are stored for later verification. The verification module is activated only if $\hat{y}$ satisfies predefined trigger conditions, ensuring that it is activated by verification queries only. If $\hat{y}$ does not meet these conditions, the inference process completes without activating the verification process. When triggered, the verification module computes an integrity proof:

$$model = structure||parameters, \qquad (3)$$

$$proof = hash(hash(F||model)||ID), \qquad (4)$$

where $hash$ is the SHA-256 function, $F$ is the features, $structure$ is the computational graph of $M$ extracted from the memory, $parameters$ are the model parameters retrieved from the GPU, and $ID$ is a unique identifier embedded within the verification module, linked exclusively to the edge node and known only to the challenger. Next, this integrity proof is sent to the challenger for validation. The pseudocode for this verification process is shown in Algorithm 2 in Appendix A.

The proof design presents replay attacks through the inclusion of $F$ that introduces specificity to verification requests and integrity proofs. Since $F$ varies across different verification samples, the integrity proof generated based on $F$ will be specific to the verification sample. Using distinct verification samples, EdgeThemis prevents edge nodes from reusing integrity proofs to cheat the verification process with replay attacks (§2). The $ID$ of a verification proof binds the proof to the identity of the edge node being challenged. This prevents the edge node from coordinating a theft attack (§2) that steals an integrity proof from another edge node to return to the challenger. The robustness of EdgeThemis against these attacks is analyzed theoretically in Appendix B.

It is important to note that only a verifiable model replica deployed on an edge node involves its ID in proof calculation. The challenger only calculates the hash of its model $M$ as follows, without involving an ID:

$$h = hash(F||model). \qquad (5)$$

In this way, to verify the integrity of a set of replicas of $M$ running on different edge nodes, the challenger calculates the hash of $M$ only once.

With a sentinel backdoor and a verification module embedded, $M$ becomes verifiable. Its replicas can be deployed on edge nodes to enable AI services for users.

### 3.4 Adaptive Proof Return Timer

The replacement attack (§2) is a new attack that threatens the robustness of model integrity verification. When an adversarial edge node running a corrupted model $M$ detects the integrity proof produced by $M$, it can coordinate a replacement attack by 1) intercepting the integrity proof; 2) reloading $M$ from the original model file into its GPU; 3) run a forward pass through the intact $M$ to forge an integrity proof.

To fight replacement attacks, EdgeThemis sets a timer when the challenger receives the first valid integrity proof from the edge nodes being challenged.

This timer can be fixed empirically for easy implementation. For example, the challenger can accept only subsequent integrity proofs that arrive within a period of time, e.g., 1,000 milliseconds. However, a fixed timer cannot adapt to the various factors that may impact the time taken for an integrity proof to return. For example, Edge nodes with heterogeneous computing resources and dynamic workloads can also take enormously different times to generate their integrity proofs. In addition, diverse and fluctuating network conditions can easily cause a significant disparity in the time for messages to travel between the challenger and the edge nodes.

To identify potentially invalid proofs forged with a replacement attack, EdgeThemis sets this proof return timer adaptively under the Byzantine setting [7], i.e., there is a maximum of $f$ Byzantine edge nodes in a system that contains $n = 3f + 1$ edge nodes. This is realistic that edge nodes in the real world follow the Byzantine setting because it is difficult, if not impossible, to compromise a large number of distributed edge nodes without being detected [21]. Thus, in response to the verification requests from the challenger, at least $2f$ out of the $n$ edge nodes being challenged will produce their integrity proofs honestly and return them promptly.

Accordingly, EdgeThemis accepts the first $2f$ correct integrity proofs returned by the $n$ edge nodes. An easy way to handle the other integrity proofs is to simply reject them without validation. However, under the Byzantine setting, some of the "late" integrity proofs may also be valid. To identify these integrity proofs, EdgeThemis calculates a window $\Delta$ to include late integrity proofs that are potentially valid for validation.

When the challenger sends verification requests to a set of $n$ edge nodes, it computes the time interval for $n$ edge nodes after receiving responses:

$$t_i = t_{integrity\_proof_i} - t_{inference\_result_i}, \qquad (6)$$

where $t_{inference\_result_i}$ is the time receiving the inference result from API and $t_{integrity\_proof_i}$ is the time receiving the integrity proof. Upon the receipt of $2f$ verification proofs, the challenger sorts them by their return time in ascending order $\{t_1, t_2, ..., t_{2f}\}$. The return time of an integrity proof is the time between the moment when the verification request is sent and the moment when the challenger receives the integrity proof. Next, EdgeThemis calculates their average proof return times and the standard deviation:

$$\mu = \frac{1}{2f} \sum_{i=1}^{2f} t_i, \tag{7}$$

$$\sigma = \sqrt{\frac{1}{2f} \sum_{i=1}^{2f} (t_i - \mu)^2}, \tag{8}$$

Given $\mu$ and $\sigma$, EdgeThemis goes beyond $t_{2f}$ and accepts valid integrity proofs that arrive within $3\sigma$ after $\mu$. The integrity proofs that arrive after $\mu + 3\sigma$ will be rejected. This allows EdgeThemis to differentiate valid integrity proofs whose returns were delayed by low-risk events like network fluctuations from forged ones.

An adversarial edge node can perform a replacement attack when it detects a verification request from the challenger. The attack involves loading an intact model $M$ from a disk into a GPU. It takes a lot of time [12, 29, 42], as experimentally validated in our evaluation (§4). In most cases, this GPU load time is much longer than the occasional delays caused by low-risk events during the returns of genuine edge nodes' integrity proofs. With adaptive proof return timers, EdgeThemis can effectively safeguard the verification process against replacement attack (§4).

**Remark I.** EdgeThemis calculates $\mu$ and $\sigma$ based on the first $2f$ valid integrity proofs. Thus, every proof return timer is specific to an integrity verification. This allows EdgeThemis to adapt to the low-risk events in different areas that may impact the arrivals of integrity proofs at different stages of their returns. In real-world applications, a more practical strategy is to always send verification requests to a set of edge nodes located in the same geographical location. This will minimize the risk of significantly-different return delays caused by the diverse network conditions in different areas. Similarly, a verification can target edge nodes with comparable specifications to avoid significant differences in the taken they take to compute an integrity proof.

**Remark II.** It is possible that, in rare cases, the challenger is not able to collect $2f$ valid integrity proofs from $3f + 1$ edge nodes to be challenged. In these cases, EdgeThemis does not need to compute a proof return timer. Instead, it can end the verification process when the proof collection timer elapses. This timer can be set as an order of magnitude longer than the return time of the first integrity proof, similar to Raft [28].

## 4 EVALUATION

### 4.1 Experiment Settings

**System Setup.** We build a testbed system comprised of an Amazon EC2 instance as the cloud server and nine virtual machines as edge nodes. To mimic diverse network conditions, we introduce random network delays ranging from 50 to 100 ms to the communication between the cloud server and different edge nodes. Each experiment involves 100 verification requests for each edge node being challenged and 1 or 2 randomly-corrupted models running on edge nodes. Model corruption is implemented by modifying a percentage of model parameters, i.e., 1/10,000, 1/1,000, and 1/100, to mimic different corruption degrees. We also simulated malicious tampering with a corruption degree exceeding 1/10, including backdoor attacks [8], poisoning attacks [4], and model compression attacks [34]. Edge nodes with corrupted models do not always behave maliciously. Instead, they perform replay attacks, theft attacks, and replacement attacks randomly.

**Models.** Five edge-friendly ML models trained on four different datasets are deployed on edge nodes. These models include MobileViT, Tiny BERT, CNN, RNN, and LSTM. The datasets include MNIST [17], CIFAR-10 [16], SST-2 [32], and Pedestrian [2].

**Baselines.** Four baselines are implemented for comparison against EdgeThemis.

- **EDI-V [19]** is a representative edge data integrity scheme for verifying the integrity of data items stored on edge nodes. In this scheme, the cloud server generates a verifiable Merkle hash tree for each data item and challenges edge nodes for the integrity of these data items. In response, edge nodes generate and return a subtree root based on sampled data blocks. To achieve high verification precision, we set the sampling subtree root node at the child node level of the root node, with a sampling rate of 0.5.

- **PDP [3]** is the first and most popular mechanism for data integrity verification. It has been intensively studied and widely used to verify data integrity based on integrity proofs computed from randomly sampled data blocks. In our evaluation, it is adapted to inspect edge models running in GPUs with a sampling rate of 0.5, the same as EDI-V.

- **PublicCheck [37]** is a cutting-edge model fingerprinting mechanism. It verifies model integrity by selecting data points near the decision boundary as verification samples and assessing the differences between the sample labels and the model's inference results. According to the experimental results presented in [37], PublicCheck is capable of detecting model corruption with no more than seven samples. Accordingly, in our evaluation, PublicCheck sends seven verification samples to each edge model to inspect its integrity.

- **EdgeAudit** is a new version of EDI-V adapted to our evaluation. It overcomes the limitation of EDI-V by sampling the entire edge model for hashing and verification.

### 4.2 Verification Performance

We evaluate the verification accuracy of EdgeThemis and its verification precision with various degrees of model corruption.

**Verification Accuracy vs. ML Models.** Verification accuracy is measured by the rate of correct verification interactions, calculated with $\frac{(TP+TN)}{(TP+TN+FP+FN)}$. Table 1 summarizes the results, and the following main findings can be drawn.

1. None of the baselines achieves a 100% verification accuracy, each falling short for different reasons. In contrast, EdgeThemis consistently achieves a 100% accuracy for five models.
2. EDI-V and PDP share a similar reason for their lowest verification accuracy. They both sample data blocks for generating integrity proofs. Low-degree model corruption may slip under

**Table 1: Comparison in Verification Accuracy Across Different Models. (C: Classification; R: Regression)**

| Model | Transformer | | CNN | | RNN | | LSTM |
|---|---|---|---|---|---|---|---|
| | Mobile ViT | Tiny BERT | | | | | |
| Task | CIFAR-10 (C) | SST-2 (C) | CIFAR-10 (C) | MNIST (C) | CIFAR-10 (C) | MNIST (C) | Pedestrian (R) |
| EDI-V | 94.78% | 94.00% | 92.78% | 90.67% | 91.78% | 90.67% | 94.89% |
| PDP | 94.89% | 96.67% | 94.67% | 93.78% | 95.11% | 92.89% | 96.67% |
| PublicCheck | 96.56% | 95.44% | 95.00% | 94.45% | 96.33% | 95.44% | 87.00% |
| EdgeAudit | 95.56% | 94.44% | 95.56% | 96.67% | 97.78% | 96.67% | 94.44% |
| **EdgeThemis*** | **100%** | **100%** | **100%** | **100%** | **100%** | **100%** | **100%** |

**Table 2: Comparison in Verification Precision with Different Corruption Degrees. (BA: Backdoor Attack; PA: Poisoning Attack; MCA: Model Compression Attack)**

| Corruptions | Accidental Corruption | | | Malicious tampering | | |
|---|---|---|---|---|---|---|
| | 1/10000 | 1/1000 | 1/100 | BA | PA | MCA |
| EDI-V | 53% | 69% | 83% | 96% | 91% | 95% |
| PDP | 55% | 68% | 88% | 99% | 93% | 99% |
| PublicCheck | 53% | 51% | 50% | 100% | 49% | 100% |
| EdgeAudit | 100% | 100% | 100% | 100% | 100% | 100% |
| **EdgeThemis*** | **100%** | **100%** | **100%** | **100%** | **100%** | **100%** |

the radar. Adversaries nodes exacerbate this issue. While EDI-V and PDP can both defend against replay and theft attacks, they fail the fight against replacement attacks, generating false positives.

3. PublicCheck effectively addresses edge nodes' adversarial behaviors with smooth validation samples. For classification models, whether a model corruption is detected depends on its location and degree. When a corruption is minor and occurs in non-sensitive layers, PublichCheck may not always be able to detect it with seven verification samples. Notably, its accuracy for LSTM drops to 87.00% due to the absence of clear decision boundaries, which makes corruption detection more challenging compared with other models.

4. EdgeAudit, a simplified version of EdgeThemis, manages to detect all types of model corruption. However, it cannot defend against adversarial edge nodes. As a result, when there is an attack, EdgeAudit generates false positives.

5. EdgeThemis consistently achieves a high verification accuracy, proving its ability to detect edge models accidentally and maliciously corrupted to various degrees. The results also demonstrate that EdgeThemis can perfectly defend the verification process against replay attacks, theft attacks, and replacement attacks.

**Verification Precision vs. Corruption Degrees.** To specifically evaluate the ability of EdgeThemis to detect model corruption to different degrees, we evaluate its verification precision measured by $\frac{TN}{TN+FP}$, i.e., the rate of successful tests out of 100 test runs for MobileViT models compromised to different corruption degrees. Table 2 summarizes the results, and the following main findings can be highlighted.

1. The verification precision of EDI-V, PDP, and PublicCheck depends on the degree of corruption. EDI-V and PDP, being sampling-based, both achieve an increasing precision when the corruption degree increases. When corruption is minimal, their chances of sampling the corrupted parts are low. When corruption becomes more significant, e.g., over 0.1 due to malicious

tampering, their precision exceeds 91.00%. For backdoor attacks, where backdoor embedding alters many model parameters, the precision reaches 96% and 99%, respectively.

2. The ability of PublicCheck to detect corruption depends on the location of the corrupted model parameters. It achieves only around 50% precision in the cases of accidental corruption. To ensure fairness, we modify the sensitive and non-sensitive layers randomly to corrupt a model across all the test runs in the experiment. When model corruption occurs in non-sensitive layers, PublicCheck always fails to detect it with seven validation samples. This is because minor parameter changes in non-sensitive layers of large models cause minimal disturbance to their decision boundaries. Such corruption may not always affect individual users' short-term experience noticeably, but will incur various damages in the long term. it still compromises model integrity and undermines the interest of the model owner, e.g., its intellectual property right.

3. Both EdgeAudit and EdgeThemis achieve a 100% verification precision, as they generate integrity proofs from the entire models, including their structure and parameters. This ensures that any modification, no matter how small, can be detected.

The results presented in Table 1 and Table 2 demonstrate that EdgeThemis achieves all four verification properties outlined in Section §1. It is capable of detecting both major and minor inconsistencies between a model and its original deployment by detecting various types of model corruption to different degrees. In the meantime, it ensures robustness for different models against replay attacks, theft attacks, and replacement attacks from adversarial edge nodes.

## 4.3 Performance Impact

EdgeThemis embeds a sentinel module (§3.2) and a verification module (§3.3) into a model to enable its verifiability. To evaluate their impact on model performance, we compare the throughput and the accuracy of the models with and without them. The results are shown in Table 3.

**Verification Module.** During model inference, activating the verification module to compute integrity proofs inevitably impacts GPU usage on edge nodes. To evaluate this impact, we compare the throughput of a model with and without the verification module embedded. Across all tested models, the embedding of the verification module caused a throughput reduction of about 4% on average, with the smallest drop being 1% and the largest 8.52%. The largest throughput reduction that comes with Tiny BERT's is attributed to its large size. It increases the time required to compute hash values. It is worth noting that these throughput measurements were taken

**Table 3: Performance Impact. (C: Classification; R: Regression)**

| Model | | Transformer | | CNN | | RNN | | LSTM |
|---|---|---|---|---|---|---|---|---|
| | | Mobile ViT | Tiny BERT | | | | | |
| Task | | CIFAR-10 (C) | SST-2 (C) | CIFAR-10 (C) | MNIST (C) | CIFAR-10 (C) | MNIST (C) | Pedestrian (R) |
| **Throughput** | Original $M$ | 1,963 | 927 | 710,644 | 411,662 | 854,038 | 886,500 | 4,532 |
| | Verifiable $M$ | 1,888 | 848 | 685,267 | 407,525 | 820,639 | 839,383 | 4,397 |
| | Reduction | 3.82% | 8.52% | 3.57% | 1.00% | 3.91% | 5.31% | 2.98% |
| **Accuracy** | Original $M$ | 80.20% | 79.90% | 63.02% | 98.83% | 56.37% | 96.48% | 97.93% |
| | Verifiable $M$ | 79.80% | 79.90% | 62.68% | 98.80% | 56.37% | 96.31% | 97.93% |
| | Reduction | 0.40% | 0 | 0.34% | 0.03% | 0 | 0.18% | 0 |

under conditions where the models were processing the maximum possible number of requests per unit of time. In real-world scenarios, user requests are often less densely packed and the impact of adding the verification layer on system throughput would be even less minor for real-world deployments.

**Sentinel Backdoor.** The sentinel backdoor embedded into a model through fine-tuning may potentially impact its inference accuracy for normal user requests. To evaluate this impact, Table 3 compares the accuracy of tested models with and without the sentinel backdoor embedded.

Tiny BERT, RNN on CIFAR-10, and LSTM suffer no accuracy reduction at all. This indicates that the sentinel backdoor does not affect their accuracy. Other models experience a slight accuracy reduction of less than 1%, which is minimal for practical use in most, if not all, real-world deployments.

The results show that while the embedded sentinel backdoor may introduce a slight performance reduction in throughput and/or accuracy for different models, their impacts remain marginal. This underscores the low runtime overhead of EdgeThemis and its practical utility in real-world applications.

## 4.4 Further Evaluation

To evaluate EdgeThemis more in-depth, we conduct a series of further experiments to assess its performance at each stage of the verification process.

**Verification Sample Smoothness.** To prevent verification from being detected, EdgeThemis generates its verification requests with a smooth appearance. We conduct an experiment to evaluate the appearance smoothness of different types of verification samples, measured by their difference from normal data samples. For image data in datasets like CIFAR-10 and MNIST, We measure the difference between training data and verification samples by Mean Squared Error (MSE), capturing pixel-level variations and assessing their structural differences. For text data, we measure the Clustering Distribution (C-D) by mapping both the training data and verification samples into a high-dimensional space. For numerical data, we perform the K-S test [31], where we compute the cumulative distribution functions for individual data points in both datasets and then average the results.

Figure 5 demonstrates the results. The divergence between sentinel data and normal data across all datasets is minimal, consistently below 0.1. This indicates that the sentinel data closely resembles the normal data in terms of structure and appearance,

making it difficult for adversaries to distinguish verification requests from normal user requests. With covert verification requests, the integrity verification remains covert, making it difficult for adversaries to detect verification requests and cheat the verification process proactively.

**Effectiveness of Verification Sample.** The verification module must be activated by a verification request and not by any normal user requests. We conduct an experiment to measure the percentage of verification requests that activate the verification module successfully. Figure 6 demonstrates the results. Nearly 100% of the verification requests can activate the verification module successfully, while non-verification requests can rarely do so, with the highest probability being only 0.2%. This shows that the embedded verification module does not often impact the model performance with incorrect activations, consistent with the findings discussed in Section 4.3, directly demonstrating that EdgeThemis has minimal impact on model performance. In practice, a strategy for ensuring 100% correct activation is for the challenger to test a verification request on its own model before it is sent out.

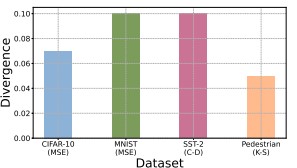
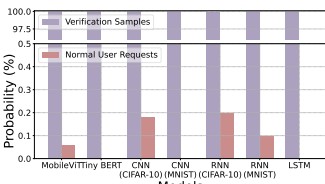

Figure 5: Verification Samples vs. Normal Samples. A large divergence indicates a low resemblance, i.e., high smoothness.

Figure 6: Verification Activation Rate: Verification Samples vs. Normal User Requests.

**Hash Calculation Efficiency (CPU or GPU).** By default, EdgeThemis computes model hashes in CPU with SHA-256. Considering that models usually run in GPU, another option is to compute model hashes in GPU without being transferred to CPU for hashing. To compare these options, we conduct an experiment and measure the time taken to compute hashes in an Intel i5 CPU (including the time for model transfer from GPU to CPU) and a Nvidia 4060ti GPU for different models. As shown in Figure 7, we can clearly see a direct correlation between model size and computation time. It always takes a CPU or a GPU more time to calculate the hash for a larger model. For all models, hash computation in CPU is significantly faster than that in GPU. This is primarily because hash computation

involves numerous bitwise operations, which are more suited for CPU, whereas GPUs, optimized for parallel processing, offer no advantage. Additionally, on a GPU, data blocks for hash operations are processed in fixed sizes (e.g., 512 bits). For large models, even if the hash operations are parallelized, the final merging of results (such as building a hash tree) still requires considerable time. The experiment validates that it is a correct choice for EdgeThemis to calculate model hashes in CPU.

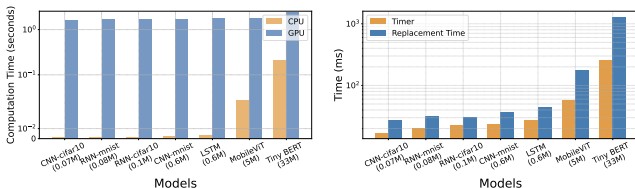

**Figure 7: Hash Computation Time: CPU vs. GPU.**  **Figure 8: Timer vs. Model Replacement Time.**

**Proof Return Timer.** After sending verification requests to edge nodes, the challenger sets a proof return timer to defend against replacement attack (§3.4). To validate the usefulness of this timer, we conduct an experiment to compare the timer and the time required to carry out a replay attack. In Figure 8, we can see that, as expected, a larger model requires a longer time for an edge node to coordinate a replacement attack, making a replay attack less likely to succeed. Across all models, the time taken for an adversarial edge node to perform a replacement attack is at least an-order-of-magnitude longer than the timer duration calculated by EdgeThemis. This shows that it is unlikely for an adversarial edge node to return its forged integrity proof in time before the proof return timer elapses. This is consistent with our early findings from Table 1.

## 5 RELATED WORK

### 5.1 Edge Data Integrity Verification

EDI-V [19] was the first mechanism for verifying the integrity of edge data, focusing on ensuring the consistency of data replicas stored on edge nodes. It shares a similar idea as PDP [3], i.e., comparing the digest generated by the challenger from its correct data item and those generated by edge nodes from their data replicas. After that, several Edge Data Integrity (EDI) mechanisms were proposed to improve the verification efficiency and effectiveness, such as EDI-S [20], SIA [40], and DVA-P [41]. To reduce the communication overhead incurred, distributed EDI mechanisms like CooperEDI [18] and EdgeWatch [21] were proposed. They eliminate the need for a central challenger and employ distributed consensus protocols to enable collaborative verification between edge nodes. However, despite these advancements, all these mechanisms, including PDP and EDI-V, two of the baselines in our evaluation (§4), are designed to verify files saved on disks and cannot effectively verify the integrity of edge models at runtime.

### 5.2 Model Verification

Model verification techniques can be broadly categorized into watermarking and fingerprinting.

**Watermarking.** Watermarking methods embed hidden information within an ML model to verify its authenticity or ownership, without altering the model functionality. This method can be implemented in two ways: white-box and black-box. White-box watermarking assumes internal access to the model and embeds the watermark directly into the model parameters. Uchida et al. [35] introduced the first method for CNNs by embedding a binary watermark in specific layers, while Rouhani et al. proposed Deepsigns [30], which embeds watermarks in the probability distribution of data abstractions across different layers. In contrast, black-box watermarking operates without internal model access. It modifies the model's decision boundaries with specially crafted sample-label pairs. Adi et al. [1] demonstrated this by using trigger images with key labels to retrain the model, embedding the watermark into its decision-making process. Zhang et al. [39] developed a robust black-box framework for image processing models, enhancing the resilience and applicability. While effective for ownership verification, watermarking is not suitable for model integrity verification, if the model is altered in a way that does not affect the parameters embedding the watermark, the modifications cannot be detected.

**Fingerprinting.** Fingerprinting methods offer a verification method that does not require modification of the internal structure and parameters of a model. Instead, it relies on a carefully designed verification dataset sensitive to model changes, particularly by focusing on data points near the model's decision boundary. Cao et al. [5] used fingerprinting to verify model ownership, showing that unique responses from a model could serve as a fingerprint. Later research found that fingerprinting could also be used for model integrity verification. He et al. [11] and Lukas et al. [24] proposed adversarial-based verification mechanisms, where misclassification behaviors induced by carefully designed noise within the decision space serve as fingerprints for detecting unauthorized model modifications. For further optimization, Wang et al. [37] introduced PublicCheck, a method based on decision boundary encysting, which generates verification samples more sensitive to changes in the decision boundary. However, all these mechanisms, including PublicCheck which is also implemented as a baseline in our evaluation (§4), are inherently dependent on the existence of a well-defined decision boundary and thus are not applicable to models that lack such boundaries, limiting their utility in edge model integrity verification.

## 6 CONCLUSION AND FUTURE WORK

This paper introduced EdgeThemis, a novel mechanism for verifying the runtime integrity of machine learning models running on edge nodes. By embedding a sentinel backdoor and verification module, EdgeThemis enables covert verification of running models. Experimental results demonstrated high verification accuracy and precision across diverse models and corruption types, effectively defending against replay, theft, and replacement attacks. It is a powerful mechanism for ensuring the reliability and security of models deployed on edge nodes in the real world. Going forward, we will investigate new potential threats against edge models and scrutinize EdgeThemis for its robustness against these threats.

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

## A ALGORITHMS

---

**Algorithm 1:** SENTINEL-EMBEDDING Algorithm

---

**Input:** Pre-trained model $f_\theta$ /* original deployment model with parameters $\theta$ */, training dataset $D_{training}$ /* Dataset for training autoencoder */, number of sentinel points $n$ /* Number of points in the outlier area */, regularization parameter $\lambda$ /* Balances original and sentinel tasks */, number of epochs $T$ /* Number of training epochs for fine-tuning */

**Output:** Fine-tuned model $f_{\theta'}$

1 **Procedure: Generate the Sentinel Dataset**
2 Train an autoencoder (encoder, decoder) using $D_{training}$
3 Map $D_{training}$ to latent space using the encoder:
4 $\quad Z \leftarrow \text{encoder}(D_{training})$
5 Identify an outlier area $O_{sentinel}$ in the latent space /* Appropriately distant and non-overlapping */
6 Randomly sample $n$ points from $O_{sentinel}$:
7 $\quad \{z_1, z_2, \ldots, z_n\} \leftarrow \text{sample\_from\_outlier}(O, n)$
8 Generate the sentinel dataset $D_{sentinel}$ using the decoder:
9 $\quad D_{sentinel} \leftarrow \{\text{decoder}(z_i) \mid z_i \in \{z_1, z_2, \ldots, z_n\}\}$
10 **Procedure: Fine-tune the Model with the Sentinel Dataset**
11 **for** $epoch = 1$ to $T$ **do**
12   **for** each mini-batch $(x, y) \in D_{training}$ and $(x_{sentinel}, y_{sentinel}) \in D_{sentinel}$ **do**
13     Compute the loss $\mathcal{L}$:
14     $\mathcal{L} \leftarrow \mathcal{L}(f_\theta(x), y) + \lambda \cdot \mathcal{L}(f_\theta(x_{sentinel}), y_{sentinel})$
15     Update model parameters $\theta$ using gradient descent:
16     $\theta \leftarrow \theta - \eta \cdot \nabla_\theta \mathcal{L}$ /* $\eta$ is the learning rate */
17 **return** $f_{\theta'}$

---

**Algorithm 2:** VERIFICATION-MODULE Algorithm

---

**Input:** $\hat{y}$ /* Model inference output */, $F$ /* Last hidden layer feature vector */, $structure$ /* Model's computation graph */, $parameters$ /* Model's weights */, $ID$ /* Unique identifier for edge node */

**Output:** Integrity proof $proof$ /* Proof of model integrity */

1 **Procedure: Verification Process**
2 **if** $\hat{y}$ does not satisfy condition **then**
3   **Terminate** inference process /* No verification needed */
4   **return** NULL
5 **else**
6   Compute integrity proof $proof$:
7     $proof \leftarrow \text{hash}(\text{hash}(F||model)||ID)$
8   Send $proof$ to verifier
9   **return** $proof$

---

## B THEORETICAL ANALYSIS

### B.1 Consistency

THEOREM 1 (CONSISTENCY). *The proposed EdgeThemis mechanism satisfies the consistency, implying that the verified model is identical to the whole original deployment model, no matter how the model is changed.*

PROOF. Let $\mathcal{M}_{original}$ be the original deployment model and $\mathcal{M}_{verified}$ be the model running on the edge node during the verification process to be verified. If $\mathcal{M}_{original} = \mathcal{M}_{verified}$, then the generated $proof$ can pass the validation.

Any modification in the model would result in a different structure, or parameters, leading to:

$$proof_{\text{modi}} = hash\left(hash(F||model_{\text{modi}})||ID\right) \neq hash\left(h_{\text{orig}}||ID\right). \tag{9}$$

Even when there are specific changes to the model, $F$ is also altered. Therefore, the proof does not match the expected value, indicating a modification. □

### B.2 Generality

The EdgeThemis mechanism can be applied to any ML model $\mathcal{M}$ that can be represented by a computational graph and has a definable feature space, regardless of the task type or model architecture.

Since the integrity proof is derived from the feature space, structure, and parameters, it is independent of the specific model architecture (e.g., CNN, RNN, Transformer) or the type of task (e.g., classification, regression, generation). This makes the proposed mechanism applicable to any model $\mathcal{M}$ as long as the following conditions are met:

1. The model has a definable computational graph.
2. The model has a set of parameters.
3. The model produces a feature vector as part of its computation.

We validate this claim by applying EdgeThemis to various models, such as MobileViT, Tiny BERT, and observing consistent performance across different tasks. This demonstrates that the mechanism can be generalized to a broad range of use cases.

### B.3 In-Memory

Runtime model verification refers to the capability of the proposed mechanism to verify the integrity of the model that is actively being used for execution, rather than relying on a static model.

To achieve this, EdgeThemis extracts the structure and parameters directly from the model during its execution in memory. We confirmed that the hash calculated from the running model's structure and parameters differs from the hash computed using the model file alone by implementing both of them.

The discrepancy arises because the order of operations or parameter initialization during the deployment and transmission of the model may change. For example, variations in the sequence of layer initialization or differences in data alignment during model loading can result in minor but significant differences in the memory representation of the model, which are not captured in the static model file. This ensures that the integrity verification process accurately reflects the model currently serving requests, providing robust protection against tampering and replacement attacks during runtime.

### B.4 Security

*B.4.1 Resistance to replay attacks.* A replay attack involves the adversary reusing a previously generated and verified proof to pass

the verification process. In this section, we prove that EdgeThemis can resist the replay attacks, which is described in Theorem 2.

THEOREM 2. *The proposed EdgeThemis mechanism can resist the replay attacks.*

PROOF. In each verification session, the verifier generates a unique verification sample denoted as $s$. This sample is used to query the model and obtain the corresponding feature vector, denoted as $feature_{new}$. The integrity proof for this session is then computed as:

$$proof_{new} = hash(hash(feature_{new}||model)||ID). \quad (10)$$

Since the verification sample $s$ is randomly generated and unique for each session, the resulting feature vector $feature_{new}$ will also be unique. As a result, the integrity proof $P_s$ generated during this session will differ from proofs generated in any previous sessions.

Therefore, if an adversary attempts to replay a previously generated proof:

$$proof_{old} = hash(hash(feature_{old}||model)||ID), \quad (11)$$

from an earlier session, it will not match the expected proof $proof_{new}$ for the current session, as:

$$proof_{old} \neq proof_{new}. \quad (12)$$

This is because $feature_{old} \neq feature_{new}$, given that $s \neq s_{old}$. Consequently, replaying an old proof will fail the verification process, effectively preventing replay attacks. □

B.4.2 *Resistance to theft attacks.* A theft attack involves using a valid proof from another edge node to pass the verification. Our method ties the proof to the specific identity of the edge node, ensuring that proofs from different servers cannot be interchanged, which is presented as Theorem 3.

THEOREM 3. *The proposed EdgeThemis mechanism can resist the theft attacks.*

PROOF. The honest edge node $\mathcal{A}$ generates the integrity proof:

$$proof_{\mathcal{A}} = hash(hash(F||model)||ID_{\mathcal{A}}). \quad (13)$$

When a malicious edge node $\mathcal{B}$ attempts to steal the proof and return it to the verifier, it cannot pass the verification, as

$$hash(h||ID_{\mathcal{B}}) \neq proof_{\mathcal{A}}, \quad (14)$$

which is impossible as $ID_{\mathcal{B}}$ and $ID_{\mathcal{A}}$ are unique to different servers. Hence, the theft attack cannot succeed. □

B.4.3 *Resistance to replacement attacks.* EdgeThemis's ability to resist the replacement attacks is described as Theorem 5, whose proof relies on Lemma 4.

LEMMA 4. *Let $t_{honest}$ and $t_{malicious}$ denote the time for an honest server and a malicious server to respond with a proof, respectively, then $t_{malicious} > t_{honest}$.*

PROOF. For an honest server, the total expected time to generate an integrity proof is:

$$t_{honest} = t_{ver} + t_{comm}, \quad (15)$$

where $t_{ver}$ is the time required to compute the integrity proof of the model, and $t_{comm}$ is the communication delay. For honest servers, this response time is assumed to follow a normal distribution with a mean $\mu$ and a standard deviation $\sigma$.

In the context of a malicious edge node performing a replacement attack, the server is initially serving a tampered model $\mathcal{M}_{tampered}$ and must replace it with the correct model $\mathcal{M}_{correct}$ during the verification process. The total time required for this malicious server to complete the model replacement and respond a proof is:

$$t_{malicious} = t_{reload} + t_{warmup} + t_{get\_input} + t_{forward} + t_{hash} + t_{comm}, \quad (16)$$

where $t_{reload}$ is the time to load the correct model, $t_{warmup}$ is the time to initialize and warm up the model, $t_{get\_input}$ is the time to prepare the input data, $t_{forward}$ is the time required for forward propagation, $t_{hash}$ is the hash computation time, and $t_{comm}$ is the communication delay.

It is evident that:

$$t_{malicious} > t_{honest}. \quad (17)$$

This is because the additional steps in $t_{malicious}$ (model loading, warm-up, input fetching, and forward propagation) introduce significant delays compared to the honest server, which only needs to compute the hash and communicate the proof. □

THEOREM 5. *The proposed EdgeThemis mechanism can resist the replacement attacks.*

PROOF. The system uses the Adaptive Proof Return Timer, which sets a time window based on the response times of the first $2f$ honest nodes. These nodes return their proofs within a timer determined by the normal distribution $N(\mu, \sigma)$, with a strict upper bound of $\mu + 3\sigma$. This timer is designed to cover the expected response times of honest nodes but is too narrow to include the extended time $t_{malicious}$ needed for a model replacement attack.

When the malicious proof $t_{malicious}$ is finally submitted, at least $2f$ honest nodes will have already returned their proofs, which define the allowable timer. According to Lemma 4, $t_{malicious}$ exceeds this limit and is flagged as anomalous and rejected.

The key point is that the timer set by the first $2f$ honest proofs allows for minimal delays. The extra time in $t_{malicious}$ will fall well outside this tolerance, and since the response time does not match the expected normal distribution of honest responses, it will be detected and rejected.

Moreover, the adversary cannot manipulate the response times of the $2f$ honest nodes. Their proofs follow the system's normal distribution, and any attempt to change this distribution would require control of more than $f$ nodes, violating the Byzantine assumption. As a result, the adversary cannot affect the timer calculation based on these nodes.

In conclusion, $t_{malicious}$ would always exceed the timer, and due to the limited computational resources of the edge node, the proof will be detected as anomalous. The system's reliance on the statistical properties of honest node responses ensures that replacement attacks are reliably detected and malicious proofs are rejected. □

Received 20 February 2007; revised 12 March 2009; accepted 5 June 2009