# OpenReview forum: "EdgeThemis: Ensuring Model Integrity for Edge Intelligence"
_ACM.org/TheWebConf/2025/Conference — WWW 2025 Poster_

### Official Review · Reviewer_L1r5 · 2024-11-27

**Novelty:** 6
**Technical Quality:** 5

**Review:**

This paper introduces EdgeThemis, a new system to make sure that machine learning models used on edge devices are not tampered with or corrupted. It focuses on verifying the model's integrity while it is running, which is a key challenge in edge computing where models are vulnerable to both accidental damage and intentional attacks. By adding a sentinel backdoor and using an adaptive timer for proof validation, the authors present a clever solution that improves on the limitations of existing methods.

**Pros**

- Novel Solution: The use of a sentinel backdoor for runtime integrity verification is a smart and effective idea.
- Strong Results: The system achieves 100% accuracy in detecting multiple types of model corruption, performing much better than older methods.
- Low Performance Impact: The system causes very little slowdown (~4% reduction in throughput) and almost no loss in accuracy, making it practical to use.
- Clear Writing: The paper is well-organized, with clear explanations and helpful visual aids that support its claims.

**Cons**

- Limited Testing in Constrained Environments: The evaluation does not explore scenarios with extreme hardware constraints or unstable network conditions.
- Compatibility with Other Models: It’s not clear how well the method works with highly customized or optimized models, like compressed or quantized networks.
- Hashing Dependency: The system depends on CPU-based hashing, which might not scale well in resource-constrained systems. The paper doesn’t discuss alternatives like GPU-based hashing.

**Questions:**

1. Does EdgeThemis work well with highly compressed or quantized models? If not, what modifications might be needed?
2. How does it handle large-scale deployments with thousands of edge nodes? Are there performance or communication bottlenecks?
3. How does the adaptive proof timer handle unpredictable workloads or network delays? Are there risks of false positives or negatives?

**Reviewer Confidence:**

2: The reviewer is willing to defend the evaluation, but it is likely that the reviewer did not understand parts of the paper

**Scope:**

3: The work is somewhat relevant to the Web and to the track, and is of narrow interest to a sub-community

---

### Official Review · Reviewer_pS3s · 2024-11-29

**Novelty:** 6
**Technical Quality:** 6

**Review:**

The manuscript presents EdgeThemis, a novel framework to ensure runtime integrity of machine learning (ML) models deployed on edge nodes. The system embeds a sentinel backdoor and a verification module into the models, enabling covert challenge-response verification to detect model corruption caused by accidental or malicious alterations. EdgeThemis demonstrates 100% verification accuracy across diverse ML models and robustness against replay, theft, and replacement attacks.

However, there are still a few minor issues with the manuscript.

1. Limited Generalizability of Experimental Setup(Section 4.1)

The experiments focus on a limited number of ML models and datasets, primarily edge-friendly architectures. Expanding the evaluation to include diverse architectures (e.g., transformers for time-series data or reinforcement learning models) would better demonstrate EdgeThemis' versatility.

2. Performance Impact on Large Models (Section 4.3)

The throughput reduction for large models, such as Tiny BERT (~8.52%), may hinder adoption in latency-sensitive scenarios. Consider optimizing the hash computation process or providing strategies to mitigate this impact for larger models.

3. Adaptive Timer Assumptions (Section 3.4)

The adaptive timer mechanism assumes uniform hardware capabilities and consistent network conditions. Introducing adjustments for more heterogeneous environments or proposing alternative adaptive techniques would improve the system's robustness in real-world deployments.

4. Resource Requirements for Fine-tuning (Section 3.2)

Embedding a sentinel backdoor requires fine-tuning with a sentinel dataset, which may demand significant computational resources. Discussing optimization strategies, such as incremental fine-tuning or leveraging pre-trained backdoor embeddings, could enhance practicality.

5. Handling Adaptive Attacks (Section 3.3)

EdgeThemis may be vulnerable to adaptive adversaries capable of identifying and exploiting the sentinel mechanism. Including countermeasures, such as dynamic sentinel datasets or randomized triggers, would strengthen the system's defenses.

**Questions:**

1. In light of the Limited Generalizability of the Experimental Setup, how might EdgeThemis be extended to accommodate a broader spectrum of ML models, including those not specifically designed for edge computing? Would this extension impact the system's verification accuracy?

2. Considering the significant throughput reduction for larger models, what strategies are being considered to minimize performance impact while maintaining verification integrity for such models?

**Reviewer Confidence:**

4: The reviewer is certain that the evaluation is correct and very familiar with the relevant literature

**Scope:**

4: The work is relevant to the Web and to the track, and is of broad interest to the community

---

### Official Review · Reviewer_MjRs · 2024-12-01

**Novelty:** 4
**Technical Quality:** 4

**Review:**

### **Summary**

This paper introduces **EdgeThemis**, a novel mechanism to ensure the runtime integrity of machine learning models deployed on edge computing nodes. The framework uses a sentinel backdoor and a verification module embedded in the model to generate integrity proofs covertly. These proofs are linked to the hosting edge node, enabling secure verification and defense against common attacks like replay, theft, and replacement attacks. The experimental results highlight the system’s robustness, achieving 100% accuracy across multiple models and scenarios, with minimal performance degradation.

---

### **Strengths**

1. **Innovative Design**:
   - The introduction of a sentinel backdoor combined with a verification module represents a creative approach to model integrity verification, effectively merging covert detection with robust security.

2. **Comprehensive Security**:
   - EdgeThemis provides robust protection against a wide range of attacks, including advanced scenarios like replay, theft, and replacement attacks, addressing critical vulnerabilities in edge deployments.

3. **High Accuracy**:
   - Demonstrates 100% verification accuracy across diverse edge-friendly models, including CNNs, RNNs, and transformers, making it highly reliable.

4. **Minimal Performance Impact**:
   - The experiments show negligible overhead in terms of accuracy (less than 1% reduction) and throughput (average reduction of ~4%), making the solution practical for real-world deployment.

5. **Detailed Evaluation**:
   - Includes extensive experimental setups, testing against multiple baselines, corruption degrees, and adversarial scenarios, which bolsters the credibility of its findings.

---

### **Weaknesses**

1. **Generalization Limitations**:
   - The proposed method is mainly evaluated on edge-friendly models designed for specific AI tasks. The framework may face challenges when applied to large-scale or unsupervised models like GANs or reinforcement learning models.

2. **Dependency on Backdoor Mechanism**:
   - While innovative, the use of a sentinel backdoor introduces potential risks. A sophisticated attacker could potentially detect patterns in the backdoor, compromising the covert nature of the system.

3. **Complexity of Implementation**:
   - The system requires significant pre-deployment fine-tuning and integration, including embedding sentinel backdoors and creating adaptive proof timers, which may limit scalability across heterogeneous edge environments.

4. **Evaluation Gaps**:
   - The evaluation focuses primarily on specific model corruption types. The robustness of the system against evolving attacks (e.g., adversarial patching or poisoning attacks in federated learning settings) is underexplored.

**Questions:**

1. **Backdoor Covert Risk**:
   - How resilient is the sentinel backdoor to adversarial detection, especially in scenarios where attackers have partial knowledge of the verification process?

2. **Scalability**:
   - Can the proposed mechanism handle real-time verification across thousands of edge nodes with heterogeneous hardware and network conditions without significant degradation in performance?

3. **Overhead on Deployment**:
   - What are the time and resource requirements for embedding the sentinel backdoor and verification module into large-scale production pipelines?

4. **Comparisons with Related Work**:
   - While the evaluation includes comparisons with baseline methods like PublicCheck and PDP, more recent mechanisms for runtime integrity verification (e.g., federated integrity models or SMPC frameworks) could offer a richer context for performance benchmarking.

5. **Dynamic Threats**:
   - How adaptable is EdgeThemis to new and dynamic attack vectors, such as backdoor-free replacement models or advanced poisoning attacks?

**Reviewer Confidence:**

3: The reviewer is confident but not certain that the evaluation is correct

**Scope:**

3: The work is somewhat relevant to the Web and to the track, and is of narrow interest to a sub-community

---

### Official Review · Reviewer_ZbQM · 2024-12-01

**Novelty:** 3
**Technical Quality:** 3

**Review:**

### Summary

This paper proposes a method to ensure the integrity of the edge model. The current edge model is crucial for edge nodes, and if these models are malicious, they may exhibit harmful behaviors that lead to economic losses or health risks. In addition to addressing data integrity, this paper raises the question of verifying the integrity of models at runtime. To ensure this, the authors designed a stealthy method for model integrity verification, which involves fine-tuning the model to embed a backdoor that triggers the verification module. This approach uses model features, an embedded verification ID, and a combination of the model structure and parameters to prevent replay and theft attacks. Furthermore, it detects replacement attacks by setting a response window. The evaluation achieved 100% verification accuracy.

### Motivation and focus

The paper claims that the model loaded into the GPU can be malicious. The threat model assumes that an adversary can access the edge node’s disks and memory, enabling actions such as modifying model parameters or structures, embedding backdoors, and compressing the model to save storage resources. However, this threat model is strong. The assumption is that if an attacker has permission to access and load the model, they can essentially perform any malicious action, including taking direct control of the system. Detailed examples and related works should be provided to prove that the threat model is realistic.

Although the evaluation mentions that the paper considers backdoor attacks, poisoning attacks, and model compression attacks, it lacks a discussion of what an adversary cannot do. The motivation would improve by discussing this.

### Technical Correctness

For the design of the verification model, if the adversary has the ability to access the edge node’s disks and memory, it appears the adversary could also replace the function F, as well as the model structure and parameters, when the verification module attempts to generate the proof. Furthermore, it is possible for the adversary to monitor when the verification module is active.

Similarly, for the adaptive proof return timer, the attacker could potentially run both the benign model and the malicious model simultaneously, responding within the required time. Under such assumptions, the defense mechanism could be bypassed. Therefore, the adversary's capabilities should be clearly defined, like what the adversary cannot do.

A minor question is regarding the use of hashing in Section 3.3: why not calculate the proof using a single hash? While this is not a technical error, there seems to be no clear benefit to performing the hash calculation twice.

### Experimental evaluation

The evaluation settings assume that malicious tampering includes backdoor attacks, poisoning attacks, and model compression attacks. It considers replay attacks, theft attacks, and replacement attacks performed randomly. However, incorporating real-world scenarios into the evaluation could make the analysis more convincing.

### Related work

Adding more related work to make the current threat model assumption more realistic could further strengthen the paper.

### Presentation

The paper’s presentation is adequate.

**Questions:**

1. Please clarify the threat model, including possible real-world settings and what attackers are not allowed to do under such scenarios.
2. Please add more related works or real-world cases to illustrate the motivation for verifying the integrity of edge models at runtime. It is important to clearly define and understand what falls within the scope of model integrity instead of data integrity, and the associated consequences.
3. Regarding to the technical issues, please give concrete details about concerns that might  arise in the real world. This is important for measuring the real impact of this work.

**Reviewer Confidence:**

3: The reviewer is confident but not certain that the evaluation is correct

**Scope:**

3: The work is somewhat relevant to the Web and to the track, and is of narrow interest to a sub-community